# Evaluating the Quality of AI-Generated Resolutions from Conversational vs Structured Sources: Implications for Enterprise Knowledge Automation

**Archan Dutta   Vinay Raj Sisodiya   Hardik Airen   Phani Nivarthi**
AI/ML Department
Aisera
{archan.dutta, vinay.sisodiya, hardik.airen, phani.nivarthi}@aisera.com

## Abstract

Enterprises increasingly rely on historical data to extract resolutions for problems and to automate knowledge mining. While structured ticketing systems (such as ServiceNow, Freshservice, Zendesk etc.) are well-established sources for resolutions, conversational platforms like Slack also capture valuable knowledge in less formal contexts. This paper proposes a Resolution Extraction System to extract meaningful resolutions for IT support cases from noisy, unstructured conversational platforms like Slack, MS Teams, etc. The paper then compares these AI-generated resolutions extracted from Slack conversations (RES) to AI-generated resolutions extracted from structured ticketing systems (RET). We evaluate six key performance indicators (KPIs) - context relevance, completeness, conciseness, noise, perplexity, and readability across 1,000 samples. Our results reveal systematic differences between structured and conversational sources. The analysis shows that with high-precision filtering, conversational sources can be transformed into a meaningful source of resolutions despite the challenges of building reliable enterprise knowledge systems from noisy data. Slack-based resolutions are more relevant and concise but noisier and less readable, whereas ticket-based resolutions are more structured and easier to interpret. These findings highlight the complementary role of conversational data for enterprise knowledge mining and provide guidance on integrating multiple sources into AI-driven automation for support and resolution.

## 1   Introduction

Most enterprises use historical information from structured ticketing systems to help solve new user problems that take the form of service desk tickets [15, 1]. Usually, companies focus on finding permanent resolutions, which are a set of instructions that can solve the user problem [11, 9]. Therefore, providing fast and high-quality resolutions can help boost productivity in these enterprises by reducing the mean time to resolution (MTTR). Conversational platforms (e.g., Slack, Webex, MS Teams chats) are another source of resolutions for service desk tickets. The aim of this study is to understand whether Slack conversations from specific channels could be a meaningful source of resolutions by comparing it to resolution from tickets. This matters in an enterprise context because it enables discovery of untapped knowledge and provides a foundation for scalable automated resolution extraction.

Slack, despite its informal and often noisy nature, contains rich, actionable content that can be effectively mined using AI-based resolution extraction and with the right pre-processing, the quality of resolutions extracted from Slack matches that of structured tickets. This study is directly motivated by the challenges of reliable ML under imperfect data conditions [8]. Unlike structured ticketing

systems, Slack conversations mix relevant resolutions with tangential discussions, emojis, and informal shorthand. This heterogeneity poses a major challenge for enterprise knowledge automation. Therefore, we propose and evaluate a resolution extraction pipeline that adapts large language models to transform Slack conversational data into reliable structured resolutions. This paper makes the following novel contributions:

- A system that transforms noisy, multi-turn enterprise chat threads into validated and structured resolutions by identifying high-quality conversations.
- Resolution-focused extraction from multi-turn enterprise chat threads, not just general summarization.
- Quality Scoring of resolutions using enterprise-specific KPIs.
- Creation of RES as distinct, reusable units in enterprise resolution serving pipelines.

## 2 Why are Conversational Platforms like Slack considered unreliable?

Conversations are often fragmented and noisy as participants often join or leave them, and important decisions are buried among reactions. Moreover, shorthand or emojis replace explicit statements. Unlike ticketing systems, which enforce schema, conversational threads lack standardized markers of "problem," "root cause," or "resolution," making downstream extraction ambiguous.

Illustrative examples make this contrast clearer. A Slack conversation might contain:

*"I think it was a config problem? → Yeah, changed env var, fixed → :thumbsup:"*

Whereas the equivalent ticket explicitly records:

*Problem: Service outage. Root Cause: Misconfigured environment variable. Resolution: Updated deployment configuration.*

Even advanced dialogue summarization approaches require explicit modeling of discourse relations to reconstruct coherence lost in conversational data [5]. Studies of dialogue annotation further reveal how unclear discourse and fragmented turns lead to inconsistency in identifying even basic features such as addressees [18]. Without additional pre-processing, a resolution extraction pipeline will miss critical context, propagate errors, or introduce low-confidence knowledge into enterprise automation pipelines.

Table 1: Comparison of Conversational vs Ticket platforms for resolution extraction

| Aspect | Conversational Platforms (like Slack) | Ticketing Platforms (like ServiceNow) |
|---|---|---|
| Structure | Informal, free-form, conversational; includes abbreviations, emojis, and incomplete sentences | Structured, standardized fields for problem description, resolution, status, priority |
| Information Fragmentation | Resolution often spread across multiple messages and threads; contributions from multiple users | Centralized; single authoritative resolution authored or approved by responsible agent |
| Mandatory Fields | No enforced title, description and resolution field; updates are inconsistent or optional | Field like title, description and resolution are often mandatory. Additionally. closure criteria is enforced |
| Noise | High Noise due to off-topic chatter, jokes, outdated suggestions | Low Noise: chronological updates focused on resolution |
| Ownership | Unclear; multiple users contribute; difficult to identify responsible resolver | Clear ownership - resolution attributed to specific agent or team |
| Auditability | Limited; messages may be deleted or edited, less queryable | Strong auditability; immutable logs, versioning, queryable records |
| Status | Slack conversations do not have a status that indicates if the conversation is resolved. | Tickets tend to have Resolved status. |

# 3 Related Work

Prior work shows that noise in dialogue (e.g., off-topic content, interruptions, unexpected phrasing) significantly hampers model performance unless carefully addressed [3]. Some aspects of knowledge mining from conversational platforms have also been explored. Dialogue summarization has been explored both in academic and industrial contexts. Feng et al. proposed a dialogue heterogeneous graph network enhanced with commonsense knowledge to improve summarization quality [4]. Wang et al. introduced Instructive Dialogue Summarization that tailors outputs via query prompts [22]. Other work has focused on extracting structured policies from dialogues in an unsupervised manner [21]. In enterprise settings, solutions such as Microsoft's Conversation Knowledge Mining leverage entity extraction, summarization, and RAG over chat transcripts for knowledge management [13, 14]. Our work extends this line of work by specifically focusing structured resolution extraction from conversational platforms. Moreover, we perform a systematic comparison of RES and RET, and evaluate across six KPIs using statistical rigor.

These characteristics motivate a method that mitigates unreliability to use Slack as a reliable contributor to enterprise resolution systems.

# 4 Methodology

## 4.1 What is a Resolution Extraction Model?

A Resolution Extraction Model is an AI system designed to process tickets and raw chats/conversations to extract three key components: *Problem*, *Root Cause* (the underlying reason), and *Resolution*. Both RES and RET will contain a Problem, Root Cause and Resolution.

## 4.2 Resolution Extraction Model for Conversational Data Sources like Slack

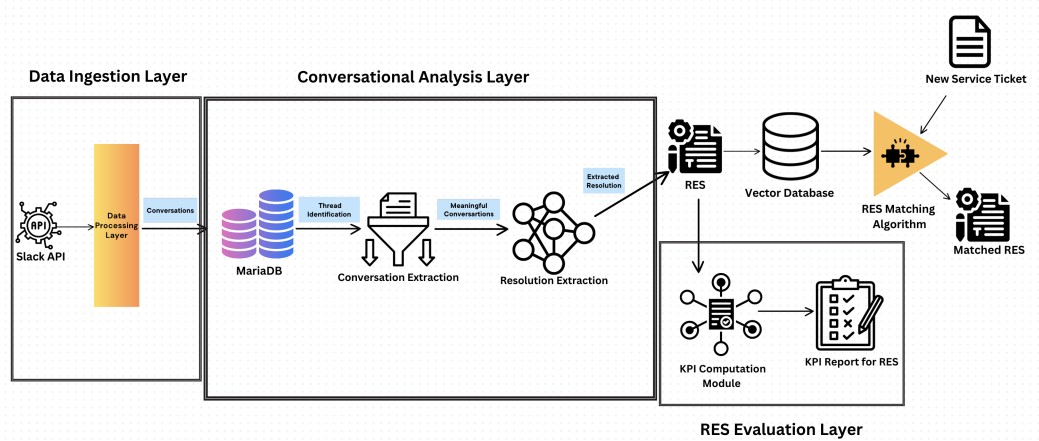

Figure 1: Resolution Extraction Pipeline from Conversational Data Source (Slack)

We propose a pipeline that adapts large language models (LLMs) to extract structured resolutions from Slack, addressing the challenge of unreliability in conversational enterprise data. The workflow is as follows:

1. **Input:** Raw Slack messages from channels via Slack Conversations API.
2. **Preprocessing:**
   - Combine related messages into coherent threads.
   - Remove Slack-specific metadata such markup tags, markdowns etc., replace IDs with names, and handle PII while retaining the textual content.
   - Preserve reactions and emojis for sentiment cues.

- Handle shorthand and informal abbreviations.

3. **Thread identification**:
   - Identify Slack messages that belong to one thread and order them chronologically to make it meaningful.

4. **Generate RES using an LLM :** Generate an RES using GPT-4o. Each RES contains the problem, the root cause and the resolution.
   - The LLM identifies resolution and creates an RES only if a concrete and complete resolution is present in the Slack thread.
   - This step acts as a crucial filtering mechanism. It is important to note that only a subset of raw threads contained a discernible resolution. For this study, we processed an initial corpus of 4500 threads to yield the 500 high-quality RES samples, resulting in a yield rate of approximately 11%. This yield rate is a key finding in itself, underscoring the necessity of the extraction pipeline to isolate actionable knowledge from conversational noise.
   - **RES Quality**: The Quality of RES has been provided in Appendix B and Appendix C.

5. **Postprocessing:** Clean and standardize the extracted outputs.

6. **Indexing:** Store RES in a vector database.

7. **Matching Algorithm:** For a new service desk ticket, find the suitable RES that will resolve the ticket.

## 4.3 Resolution Extraction Model from Tickets

The RET pipeline follows a similar process as RES pipeline with the exception of certain preprocessing steps that are specific to RES. These pipelines tend to use standard fields such as title, description and comments for resolution extraction. For this study, we processed an initial corpus of 1700 tickets to yield the 500 high-quality RET, resulting in a yield rate of approximately 30%. The yield rate is higher when compared to RES because of the structured nature of tickets. Similar to RES, we have used GPT-4o to extract resolutions from Tickets.

## 5 Experimental Setup

In our experiment, RET serves as a control, while RES represent extractions from noisy conversational data source. Our experiments test whether the proposed pipeline can extract reliable resolutions from Slack threads, and how closely these outputs approximate the quality of RET across multiple KPIs. We randomly sampled 500 RES and 500 RET outputs. Six KPIs were defined and computed for both groups, then compared using statistical tests. Each KPI is visualized using kernel density plots and analyzed by measures of central tendency. Three different LLMs: GPT-4o, o4-mini (a specialized Chain of Thought model), and Llama 3.3 70B were used to compute the KPIs. Since the initial resolution extraction was performed with GPT-4o, incorporating multiple models for evaluation helped minimize bias and ensured a more balanced and reliable judgment. Llama3.3 was included specifically to ensure that KPIs are also computed from a LLM outside of GPT family of LLMs.

### 5.1 Hardware and Software requirements
- All analysis was performed using Python libraries: `scipy`, `numpy`, `matplotlib`, `seaborn`, `pandas`, `textstat`, `transformers`, `torch`, and `openai`.
- Compute resources: The experiment runs an AWS instance - `r5.4xlarge`. It also uses `GPT-4o-2025-01-01-preview`, `o4-mini-2024-12-01-preview` hosted on Azure OpenAI, and `Llama-3.3-70B-Instruct` hosted on an AWS instance - `p5en h200`

### 5.2 KPIs Explained

Four KPIs (Context Relevance, Completeness, Conciseness, Noise) were computed using LLMs. Perplexity and Readability are non-LLM based KPIs and were computed using pretrained model (`EleutherAI/gpt-neo-1.3B`) [2, 6] and standard readability score (`Flesch-Kincaid`) [19, 7] respectively.

- **Context Relevance:** Relevance to the source ticket or Slack conversation. Higher is better.

- **Completeness:** Does the resolution fully address the issue? Higher is better.

- **Conciseness:** How succinct the resolution is without losing meaning. Higher is better.

- **Noise:** Amount of irrelevant or extraneous information. Lower is better.

- **Perplexity:** Measure of textual complexity. Lower is better.

- **Readability:** Ease of reading, favoring structured resolutions. Higher is better.

### 5.3 Statistical Tests

Slack and Ticket are completely independent data sources which means that the data distributions for RES and RET are also independent. To compare RET and RES distributions for each KPI, we applied statistical significance tests: Mann-Whitney U test [16] and Kolmogorov-Smirnov (KS) test (which does not assume normality) [20]. Furthermore, we also applied statistical tests for effect size measures: Cliff's delta, a robust nonparametric metric that is particularly suitable for nonnormal data [12, 10], and Wasserstein distance [17].

## 6 Results & Analysis

### 6.1 Significance Tests

The null hypothesis stated there would be no difference between RET and RES KPIs. For GPT-4o three out of four KPIs, p-values $< 0.05$ allowed rejection of the null hypothesis, confirming the differences in each KPI distribution. For Conciseness, Mann-Whitney $p = 0.0524$ failed to reject the null hypothesis, while KS test indicated distribution differences.

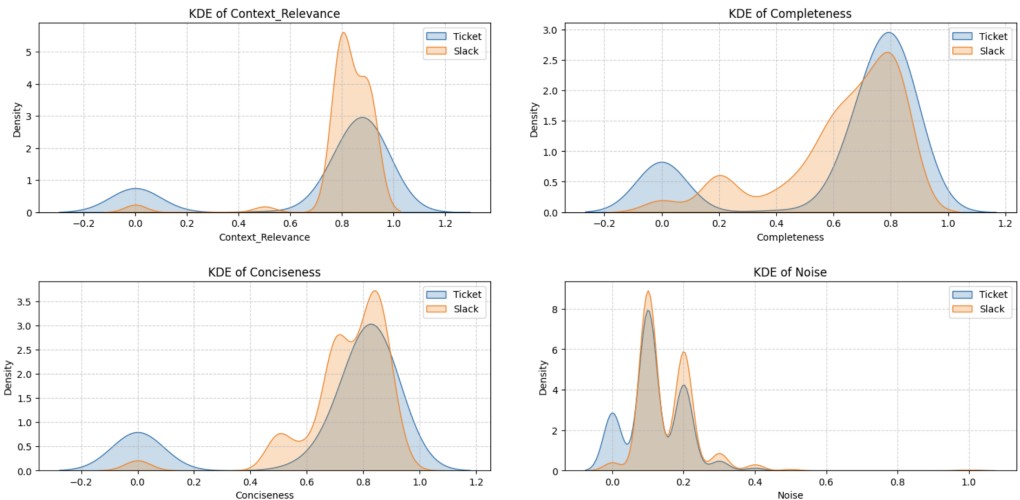

Figure 2: Kernel density plots of KPIs computed by GPT-4o

Extending the analysis to KPIs generated by o4-mini and Llama3.3, we again applied both statistical tests across all six KPIs. For o4-mini, three of the four KPIs produced statistically significant differences between RES and RET ($p < 0.05$), Unlike GPT-4o, both tests failed to reject the null hypothesis for Completeness (Mann-Whitney $p = 0.688$ while the KS test $p = 0.0534$.
For Llama3.3, all four KPIs produced statistically significant differences between RES and RET ($p < 0.05$).

This means that across GPT-4o, o4-mini and Llama3.3, the results consistently indicate that RES and RET differ for the majority of KPI distributions.

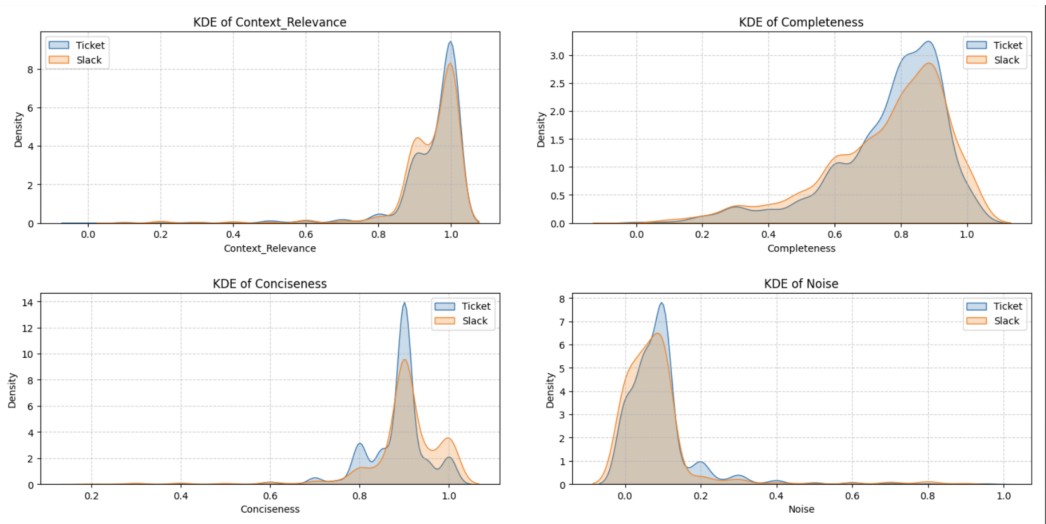

Figure 3: Kernel density plots for KPIs computed by o4-mini

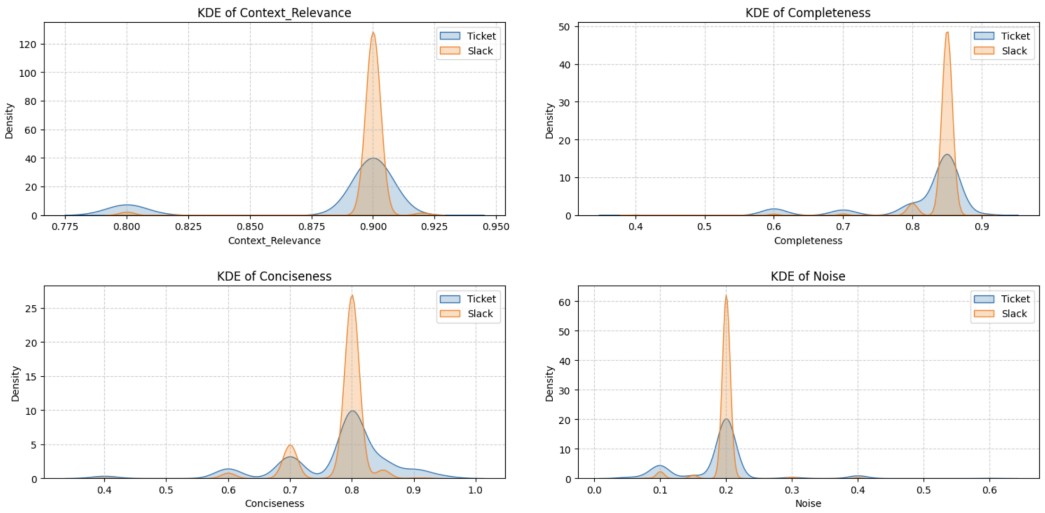

Figure 4: Kernel density plots of KPIs computed by Llama 3.3

## 6.2 Findings from LLM based KPIs

Table 2: RET vs RES - Qualitative comparison based on mean of the distribution across KPIs

| KPI | GPT-4o | o4-mini | Llama3.3 |
|---|---|---|---|
| Context Relevance | RES is better | Similar | Similar |
| Completeness | Similar | Similar | RES is better |
| Conciseness | RES is better | Similar | Similar |
| Noise | RET is better | Similar | Similar |

As judged by all LLMs, RES has nearly identical or higher Context Relevance and Conciseness scores with GPT-4o highlighting that RES is much better than RET in terms Context Relevance and Conciseness (provided in Appendix C). Cliff's Delta scores and Wasserstein distance) suggest small

effect sizes across all LLM based KPIs, indicating that these differences are present but not large (provided in Appendix B).

## 6.3 Findings from Non-LLM based KPIs

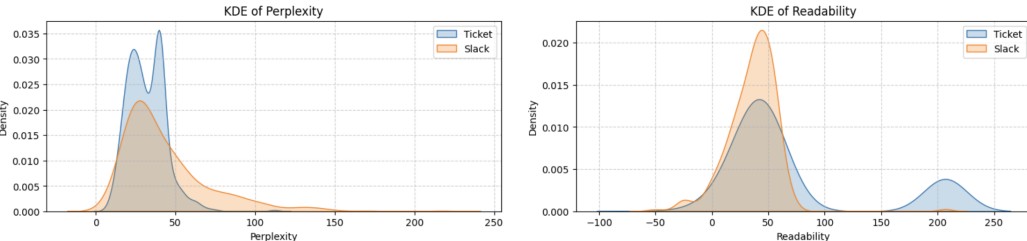

Figure 5: Kernel density plots for Perplexity and Readability for RES and RET

The two KPIs that show the most dramatic differences are Perplexity and Readability. RES has significantly higher perplexity (Wasserstein distance = 11.64), reflecting more complex and less predictable language. RET has substantially higher readability, confirmed by effect size measurements (Cliff's Delta = 0.2732 and Wasserstein distance = 34.7).

This aligns with the structured, formal nature of tickets. Based on the results, we can confirm that Readability and Perplexity are the major separator between RES and RET.

Table 3: Quantitative comparison of RET vs. RES for Perplexity and Readability.

| KPI | RET (mean) | RES (mean) | RET (std) | RES (std) |
|---|---|---|---|---|
| Perplexity | 31.98 | 43.62 | 11.813 | 27.86 |
| Readability | 70.41 | 35.71 | 66.47 | 22.92 |

## 6.4 KPI Interpretations

These findings demonstrate how Slack has unfavorable KPI values for Noise, Perplexity and Readability (probably due to off-topic chatter, informal language). Although differences are statistically significant, not all have strong practical implications. Illustrative examples make this clearer, here's a RES that has low readability and high perplexity because of unusual phrasings and less predictable langugage structure.

*Problem: Payroll tax calculation updates for Q1 are not applied automatically.*
*Root Cause: The system requires customers to manually enable the new payroll tax calculation settings.*
*Resolution:*
*1. Inform customers that Q1 payroll tax updates are not applied automatically.*
*2. Advise customers to manually enable the new calculation settings.*
*3. Include a note in the release documentation to highlight this requirement and the system's upcoming flagging of discrepancies in old settings.*

In contrast to this, here is the RET, which has high readability and low perplexity.

*Problem: Unable to update user role permissions in Admin Portal.*
*Root Cause: Mismatch in the role ID mapping for the user in the database.*
*Resolution:*
*1. Identify the affected user and confirm the issue is specific to their profile.*
*2. Attempt to remove the user from their current role, save the changes, and re-add them with updated permissions.*
*3. If the issue persists, create a new user profile for the individual and assign the updated permissions*

*to the new profile as a temporary workaround.*
*4. Wait for the engineering team to deploy a patch to fix the role ID mapping issue in the database.*
*5. Once the patch is deployed, revert to the original profile and test the changes.*

This shows that Slack's lower Readability and higher Perplexity doesn't make it unusable, it means that:

- Slack is context-rich and, therefore, requires more sophisticated pre-processing steps to remove noise.
- Better techniques for consolidating multiple threads about the same context to extract complete resolutions.

Table 4: Summary of KPI interpretations comparing RES vs RET

| KPI | Interpretation |
| --- | --- |
| Context Relevance | Slack shows better relevance. |
| Completeness | Nearly identical overall; small statistical difference but not practically significant. |
| Conciseness | Slack is more concise on average. |
| Noise | Slack has slightly more noise overall, consistent with conversational style. |
| Perplexity | Slack exhibits much more varied and complex text. |
| Readability | Tickets are significantly more readable, confirmed with effect size measures. |

## 7 Limitations

The following are the limitations of this study:

1. The prompt for LLM-based KPI computation are same across the three LLMs (GPT-4o, o4-mini, Llama3.3)
2. What constitutes a "complete resolution" may vary between users, teams, or domains, affecting consistency.
3. Model performance may vary with update to LLMs, making replication challenging if a different version is used.
4. Limited sample size of 500 for both RET and RES.

## 8 Conclusion

Our results show that a carefully designed resolution-extraction pipeline can yield knowledge artifacts from successfully filtered conversational enterprise data, making it a reliable source for automation. Moreover, the comparative analysis demonstrates that the source of resolutions - Slack or Tickets, has a statistically significant and practically relevant impact on the linguistic and structural characteristics of the extracted resolution.

- Resolutions extracted from Slack (RES) are more relevant and concise but could be less readable. It also suggests that Slack can be a really good source of information for problem resolution. Extracting Resolution from Slack requires a "High precision" extraction system to filter out noise.
- Resolutions extracted from Tickets (RET) are more structured, readable.

Given the high volume and speed of discussions that take place on collaborative platforms like Slack, this opens up promising opportunities for knowledge mining, incident resolution, and building self-service AI assistants. For enterprises, this means that valuable operational knowledge is not limited to structured ticketing systems. It is also embedded in peer-to-peer discussions on Slack (or similar platforms such as MS Teams, Cisco Webex). In doing so, enterprises can reduce MTTR. This study makes the following contributions:

1. It demonstrates how noisy, multi-turn chat threads can be transformed into validated and structured resolutions.

2. It focuses on resolution-extraction rather than general summarization.

3. It introduces quality scoring driven by enterprise-specific KPIs.

4. it defines RES as distinct, reusable units enterprise resolution pipelines.

By systematically quantifying reliability dimensions and validating with statistical tests, this work contributes practical insights into automated resolution extraction systems for noisy, real-world enterprise applications.

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

# A Glossary of Terms

Table 5: Glossary of key terms.

| Term | Definition |
| --- | --- |
| RET | Resolution Extracted from Tickets. |
| RES | Resolution Extracted from Slack. |
| Problem | The user-reported challenge described. |
| Root Cause | The underlying technical or operational reason for the issue. |
| Resolution | The fix, workaround, or action taken to address the problem. |
| Resolution Extraction Model | AI system that generates structured resolutions from text. |

# B Mean and Standard Deviations of LLM based KPIs

Table 6: Mean and Standard Deviations of KPIs computed by GPT-4o

| KPI | RET Mean | RES Mean | RET Std Dev | RES Std Dev |
| --- | --- | --- | --- | --- |
| Context_Relevance | 0.7098 | 0.8152 | 0.3408 | 0.1439 |
| Completeness | 0.6352 | 0.6296 | 0.3081 | 0.2713 |
| Conciseness | 0.6663 | 0.7400 | 0.3209 | 0.1603 |
| Noise | 0.1191 | 0.1527 | 0.0877 | 0.0873 |

Table 7: Mean and Standard Deviations of KPIs computed by o4-mini

| KPI | RET Mean | RES Mean | RET Std Dev | RES Std Dev |
| --- | --- | --- | --- | --- |
| Context_Relevance | 0.9460 | 0.9402 | 0.1061 | 0.1104 |
| Completeness | 0.7640 | 0.7552 | 0.1713 | 0.1882 |
| Conciseness | 0.8844 | 0.8985 | 0.0701 | 0.0964 |
| Noise | 0.0910 | 0.0841 | 0.0982 | 0.1219 |

Table 8: Mean and Standard Deviations of KPIs computed by Llama3.3

| KPI | RET Mean | RES Mean | RET Std Dev | RES Std Dev |
| --- | --- | --- | --- | --- |
| Context_Relevance | 0.8845 | 0.8985 | 0.0361 | 0.0130 |
| Completeness | 0.8146 | 0.8437 | 0.0760 | 0.0322 |
| Conciseness | 0.7762 | 0.7823 | 0.0920 | 0.0499 |
| Noise | 0.1877 | 0.1971 | .0618 | 0.0256 |

# C Statistical Tests and Effect Sized for all LLM based KPIs

Table 9: Statistical Tests and Effect Sized for all KPIs computed by GPT-4o

| KPI | Mann-Whitney p-value | KS Test | Cliff's Delta | Wessertein Distance |
| --- | --- | --- | --- | --- |
| Context_Relevance | 0.0091 | 0.0000 | 0.0864 | 0.1434 |
| Completeness | 0.0000 | 0.0000 | 0.1918 | 0.0944 |
| Conciseness | 0.0523 | 0.0000 | 0.0691 | 0.1129 |
| Noise | 0.0000 | 0.000007 | -0.2168 | 0.0336 |

Table 10: Statistical Tests and Effect Sized for all KPIs computed by o4-mini

| KPI | Mann-Whitney p-value | KS Test | Cliff's Delta | Wessertein Distance |
|---|---|---|---|---|
| Context_Relevance | 0.0027 | 0.0204 | 0.0577 | 0.0078 |
| Completeness | 0.6884 | 0.0534 | 0.0084 | 0.0198 |
| Conciseness | 0.0000 | 0.0000 | -0.1783 | 0.0250 |
| Noise | 0.0000 | 0.0000 | 0.1179 | 0.0180 |

Table 11: Statistical Tests and Effect Sized for all KPIs computed by Llama3.3

| KPI | Mann-Whitney p-value | KS Test | Cliff's Delta | Wessertein Distance |
|---|---|---|---|---|
| Context_Relevance | 0.0000 | 0.0000 | -0.1461 | 0.0140 |
| Completeness | 0.0000 | 0.0000 | -0.1937 | 0.0303 |
| Conciseness | 0.0107 | 0.0000 | 0.0456 | 0.0332 |
| Noise | 0.0000 | 0.0000 | -0.1370 | 0.0239 |

