# OpenReview forum: "Evaluating the Quality of AI-Generated Resolutions from Conversational vs Structured Sources: Implications for Enterprise Knowledge Automation"
_NeurIPS.cc/2025/Workshop/Reliable_ML — NeurIPS 2025 - Reliable ML Workshop_

### Official Review · Reviewer_3pe9 · 2025-09-14
**Relevant paper for Reliable ML from Unreliable Data.**

**Rating:** 7
**Confidence:** 3

**Review:**

This study asks whether conversational enterprise platforms can serve as reliable sources of IT resolutions, in contrast to structured ticketing systems. The authors process noisy, multi-conversational Slack threads to extract structured “Problem, Root Cause, Resolution” triples. They compare these to ticket derived resolutions across six KPIs. Using GPT-4o, o4-mini, Llama 3.3 and a statistical analysis, they show that Slack-derived resolutions are more contextually relevant and concise but less readable and noisier than tickets. The work is interesting in demonstrating with careful filtering, conversational data can be used for enterprise knowledge mining.

Strengths:
Relevant to the workshop theme, specifically reliable ML under imperfect/noisy data track. The focus on targeted resolution extraction instead of summarizing is a strength since the context of the original resolution is immediately available.

Weaknesses:
The replicability and reproducibility of the pipeline is not tested, is it intended to be domain specific? Can it be tested across other enterprise domains like HR, or Accounting? The study has a limited sample size and could be extened to a larger dataset of conversations. Some KPIs (like completeness) are subjective and this could affect replicability.

Notes for authors:
Include an analysis of false positives and hallucinations to understand the pipeline’s failure modes. Since slack has poor readability in its resolutions, authors should consider adding a summarization post-processing step.

---

### Official Review · Reviewer_YYNQ · 2025-09-19
**Review for Evaluating the Quality of AI-Generated Resolutions from Conversational vs Structured Sources**

**Rating:** 5
**Confidence:** 3

**Review:**

### Summary
This paper tests if useful resolutions can be extracted from Slack conversations. The authors built a system that extracts (Problem, Root Cause, Resolution) triples from slack chats, so that it can later be used for enterprise for AI/LLM-based automation. It requires careful experiment design and engineering because Slack outputs are more concise and relevant but noisier and harder to read, while ticket data is cleaner and better structured. The paper clearly demonstrate that chat platforms such as Slack can supplement ticket data to provide structured inputs for downstream enterprise automation

### Strengths
The paper as several strengths:
- Clear motivation and problem framing.
- Good structure and experimental setup
- Evaluation covers multiple metrics (relevance, completeness, conciseness, readability, perplexity) that are relevant for corporate contexts and uses several LLMs as judges.
- The study is practical and relevant to enterprise settings where data is messy and unstructured.

### Weakness/Limitations
I personally feel that the papers findings are not very surprising given current LLM capabilities. It is well known that LLMs can extract useful information from messy data and put it in a structured format.

In addition, the reliability of LLMs as evaluators is not thoroughly discussed/examined, leaving questions about how well their scores align with human judgments. Would be great if they can show some examples where their scores assigned by LLM judges, when ranked in their assigned scores, show a consistent increase in quality (or which ever criteria we are looking at).

The proprietary nature of the data and lack of code release limit the paper’s reproducibility, as the authors acknowledged.

### Suggestions
- Show evidence that LLM scores meaningfully align with human evaluations, with concrete examples.
- Table 1 is a bit hard to read, I’d recommend adding more spaces between rows, or reduce some text.